# Hyperspectral target detection based on graph sampling and aggregation network

**Tie Li**⊕, **Hongfeng Jin**⊕[ID]*, **Zhiqiu Li**⊕

School of Electronics and Information Engineering, Liaoning Technical University, Huludao, Liaoning, China

⊕ These authors contributed equally to this work.
* jinhongfeng1109@163.com

**Funding:** Scientific Research Fund of Liaoning Provincial Education Department (LJ2020JCL007).

## Abstract

To comprehensively utilize the spectral information encapsulated within hyperspectral images and more effectively handle the intricate and irregular structures among pixels in complex hyperspectral data, a novel graph sampling aggregation network model is put forward for hyperspectral target detection. Notably, before this study, graph sampling aggregation networks had scarcely been employed in the realm of hyperspectral target detection. This proposed model is capable of autonomously learning the effective feature representations of nodes within the graph, thereby facilitating the extraction and processing of graph data. It achieves this by extracting feature vectors through principal component analysis to construct adjacency matrices and performing convolution operations on hyperspectral images via sparse matrix multiplication, which enables the propagation and aggregation of node features within the graph structure. Upon reconstructing the image, the target data is extracted using residuals, and target detection is accomplished by minimizing the constraint energy. The model was evaluated on seven hyperspectral image datasets, and the experimental results demonstrated that the proposed graph sampling aggregation network model could proficiently detect targets with an average detection accuracy exceeding 99.8%, outperforming other comparative models. Concurrently, it exhibits a remarkable adaptability to the diverse characteristics of different datasets, thus validating its high level of accuracy and robustness.

## Introduction

With the continuous development of hyperspectral imaging technology, hyperspectral imaging has become an important means of obtaining spectral information about ground targets. The ultra-high spectral resolution enables imaging spectrometers to provide rich textures and details in the images obtained [1]. It has been widely used in national economic fields such as resource exploration, environmental detection, fine agriculture, and climate change [2]. Hyperspectral image (HSI) is a three-dimensional image that contains rich spectral and spatial information, with the characteristic of 'integration of spectral and spatial information', where each pixel corresponds to a spectral curve. Hyperspectral target detection uses this

massive data to detect the target of interest from the HSI, and determine its position and category [3], which plays a key role in practical applications.

Traditional detection methods have relatively fixed parameters and algorithms, making it difficult to cope with changes in target morphology, and have poor performance in dealing with irregular structures between pixels in hyperspectral data. For example, detectors based on Constrained Energy Minimization (CEM [4]) may overlook the influence of target pixels when estimating background pixels; The detector based on adaptive consistency/cosine estimator (ACE [5]) typically assumes that there is additional noise in the background but not in the target, which often does not hold in practical situations. To further investigate the intrinsic properties of HSI data, many researchers have been working to develop more complex modeling techniques, such as the E-CEM [6] detection method, which requires multiple samples with replacement and the construction of multiple detectors. When the parameters are small, the algorithm performance is not stable enough. The kernel-based OSP detector [7] maps the original space to the kernel space, allowing traditional OSP [8] methods to effectively utilize the nonlinear characteristics of HSI. However, this mapping process may result in the loss or distortion of some data information.

In recent years, deep learning has made significant progress in fields such as pattern recognition and computer vision. Due to the better expressive ability of deep spatial-spectral features [9], some scholars have used deep learning to construct target detection models and achieved significant results. Xie et al. proposed a deep learning algorithm BLTSC based on background learning [10], which requires inputting background samples into an AAE network [11] model for training. However, the AAE network structure is complex and requires a large amount of computational resources to learn the feature representation of the background samples during the training process. Meanwhile, BLTSC mainly focuses on background learning and lacks targeted mechanisms for target feature extraction, resulting in poor detection performance on certain datasets. Shen et al. proposed a detection method based on interpretable representation networks, HTD-IRN [12]. Although converting the physical model into a deep learning network achieves compatibility between nonlinear representation and physical interpretability, there are still shortcomings in target feature extraction in complex environments, making it difficult to accurately distinguish the boundaries between the target and background when reconstructing them, resulting in detection results only on some datasets. Zhou et al. proposed a detection algorithm CEM-VAE [13] based on constraint energy minimization variational autoencoder, which uses background to calculate the autocorrelation matrix and introduces CEM regularization term to preserve only background information in the reconstructed samples. However, in the case of small targets or unclear target features, it cannot effectively extract target features, resulting in poor detection performance. Tian et al. introduced the OSP concept into VAE [14] networks and proposed an orthogonal subspace-guided variational autoencoder learning method for real background representation, OS-VAE [15]. This method can train more accurate and reliable background representation models, but due to its emphasis on background suppression and representation, it may to some extent neglect the learning of target features. Li et al. proposed a hyperspectral target detection method SSROW [16] based on spatial-spectral reconstruction and operator weighting. Although it can achieve high-accuracy detection without coarse separation of target and background samples, it requires principal component analysis (PCA [17]) to extract feature vectors, multiple image processing, and operator construction, which increases computational complexity and operational difficulty.

In response to the above issues, the proposed hyperspectral target detection model SAGE based on graph sampling and aggregation networks has significant advantages. In terms of

automatic learning feature representation, the SAGE model directly inputs all data into the model and describes the relationships between pixels by constructing adjacency matrices. In graph convolution (GNN [18] [19] [20]) operation, the adjacency sparse matrix is multiplied with the node dense matrix, so that each node can fully obtain the information of its neighboring nodes. Through multiple iterations of updating node features, more representative feature representations are gradually learned. This approach differs from traditional methods and other deep learning models in that it can automatically learn complex relationships between pixels directly from image data, without the need for complex sample preprocessing, and can more accurately capture feature information of small and inconspicuous targets. Compared with other models, the SAGE model can effectively avoid the interference of irregular structures on target detection and accurately identify target pixels when processing hyperspectral data containing complex terrain and targets. In terms of improving detection accuracy and efficiency, the SAGE model reduces unnecessary computational complexity and avoids the problem of duplicate parameter settings by optimizing graph structure construction and graph convolution operations.

## The proposed model and algorithm

The constructed SAGE model is shown in Fig 1. In the preprocessing stage, the main features are extracted using PCA, while the eight neighboring pixels of each pixel are considered, and an index is constructed by an adaptive thresholding strategy and cosine similarity. The index

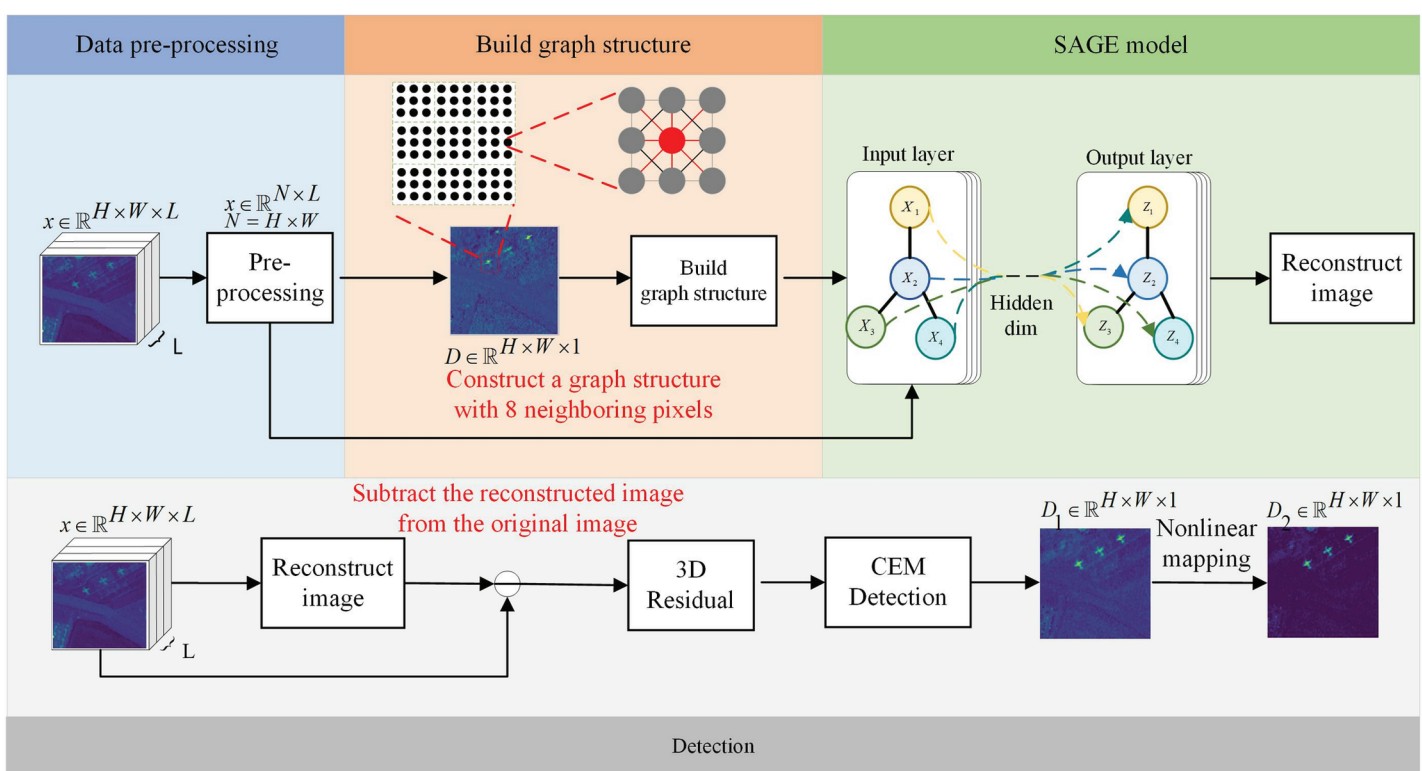

**Fig 1. Flowchart of the proposed target detection method.**

can accurately reflect the similarity and correlation between pixels, which is then used to construct the adjacency matrix to provide a structural basis for the model to learn the feature representation of nodes (i.e., pixels) in the graph. After reconstructing the image, the 3D residuals between the target and the original image are calculated to highlight the target. Finally, the residuals are detected using CEM and a nonlinear mapping function is applied to obtain results.

## Preprocessing

Suppose a given HSI is composed of $N$ pixels and $L$ bands, Denote $x = [x_1, x_2, \cdots, x_N]$, where $x_i \in \mathbb{R}^{L \times 1}$, $i = 1, 2, \cdots, N$. Given the prior target information $d \in \mathbb{R}^{L \times 1}$. CEM suppresses background and extracts targets by designing a finite impulse response linear filter. The CEM operator calculation formula is:

$$w = \frac{R^{-1}d}{d^{\mathrm{T}}R^{-1}d} \tag{1}$$

In this equation, $R = \frac{1}{N}\sum_{i=1}^{N} x(x^{\mathrm{T}})$ represents the autocorrelation matrix. Apply the CEM operator to each pixel in the image to obtain the distribution of the target in the image, achieving detection of the target. The output of the filter is the detection result:

$$\xi_{CEM} = w^{\mathrm{T}}x = \left(\frac{R^{-1}d}{d^{\mathrm{T}}R^{-1}d}\right)^{\mathrm{T}} x = \frac{d^{\mathrm{T}}R^{-1}x}{d^{\mathrm{T}}R^{-1}d} \tag{2}$$

The SAGE detection process takes the entire hyperspectral image as input and first normalizes the data to the $[0, 1]$ interval, making the data distribution more regular and helpful for model training and convergence, providing more representative inputs for the model. The normalization formula is:

$$x_n = \frac{x - \min(x)}{\max(x) - \min(x)} \tag{3}$$

In the formula, $\min(x)$ and $\max(x)$ respectively represent the minimum and maximum values in the input data $x$. In order to further improve data quality and filter out noise, feature enhancement [21] is performed on the normalized data $x_n$. First, use a Gaussian filter to smooth $x_n$ and obtain $x_n'$ to reduce the impact of noise; Multiply $x_n'$ by the adjustment parameter $\beta$ (which can be determined by using the mean of the input data), and then use the hyperbolic tangent function tanh for nonlinear transformation to obtain the enhancement coefficient $\delta$, which is used to enhance the contrast and details of the image. The specific calculation formula is:

$$\delta = \tanh(\beta x_n') \tag{4}$$

In the formula, $x_n' = GF(x_n)$, where $GF(\cdot)$ represents Gaussian filtering. Apply $\delta$ to $x_n$ to obtain the enhanced result.

## Build graph structure

To enable effective propagation of input data on the graph and better feature learning, it is necessary to construct a neighborhood matrix to determine the information propagation path

and handle the irregularity between hyperspectral pixels. This approach does not rely on traditional assumptions of regular pixel relationships and can adapt to complex spatial and spectral feature relationships between pixels. It captures pixel relationship information in irregular structures through edges in the graph structure, enabling feature propagation and aggregation based on these relationships in subsequent operations such as graph convolution. The selection of 8 neighboring pixels is based on the consideration of pixel spatial correlation, which can better capture local structural information. The setting of the spectral cosine similarity threshold can effectively connect similar pixels and construct representative graph structures. The formula for calculating cosine similarity is:

$$\phi = \frac{\sum x_b y_b}{norm_x \times norm_y} \tag{5}$$

In this equation, $x_b$ represents the current pixel, $y_b$ represents neighboring pixels, $norm_x = \sqrt{\sum x_b^2}$, $norm_y = \sqrt{\sum y_b^2}$.

The specific method is as follows: Firstly, PCA is used to perform dimensionality transformation and noise filtering on the feature-enhanced results to obtain b, and based on this, the total number of pixels $N$ is obtained. Next, traverse each pixel, determine its surrounding 8 neighboring pixels, calculate the coordinates of these neighboring pixels, and ensure that the neighboring pixels are within the image range and different from the current pixel. When constructing the edge index of the graph, 8 neighboring pixels provide specific candidate neighbors. Calculate the cosine similarity between the feature vectors of each pixel and its neighboring pixels after PCA processing. If the cosine similarity is greater than the given $\rho$(selecting 75% of all similarity lists as the threshold), then these two pixels are considered similar, and the index of the neighboring pixel is added to the corresponding pixel's neighboring pixel list. Subsequently, for each neighboring pixel in the neighboring pixel list, the index pairs of the current pixel and neighboring pixels, as well as the index pairs of neighboring pixels and current pixels, are added to the edge list to construct the edges of the undirected graph. Finally, an adjacency matrix is constructed using the edge list and the total number of pixels as inputs to complete the construction of the graph structure. The specific adjacency matrix diagram is shown in Fig 2.

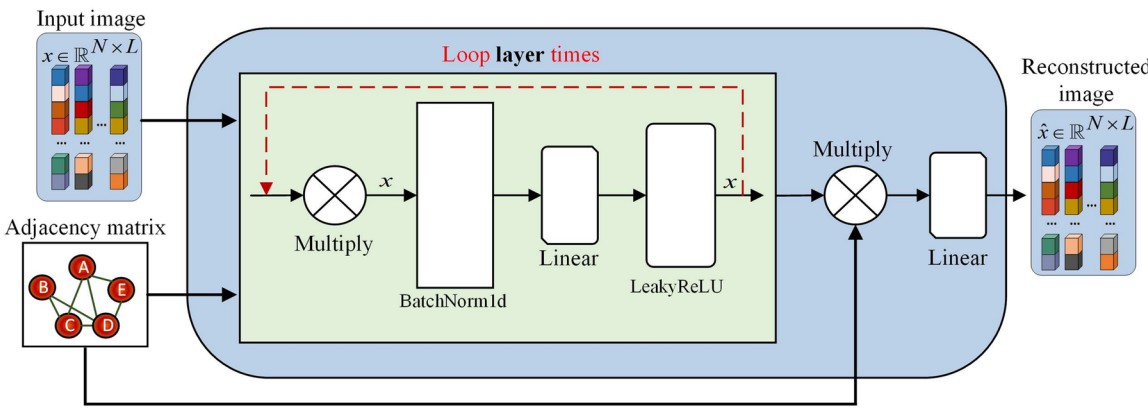

**Fig 2. SAGE model flowchart.**

## Graph sampling aggregation network model

The SAGE model mainly consists of multiple linear layers (Linear), batch normalization layers (BatchNorm1d), and activation functions (LeakyReLU [22]). It implements graph convolution function through sparse multiplication operation (Multiply), which is equivalent to propagating information on the graph, enabling the model to adapt to the complex structure and feature distribution of hyperspectral images. In the process of model training, in order to better extract image features, the following two loss functions were used: the mean square error loss was used to measure the difference between the predicted and true values of the model, enabling the model to better fit the data; Use L2 regularization [23] to control the complexity of the model and prevent overfitting during model training. Combining the two as the loss function of the model, the specific formula is as follows:

$$loss = 0.1 \times \frac{1}{n} \sum_{i=1}^{n} (b - \hat{b})^2 +$$
$$0.001 \times \frac{1}{2} \sum_{i=1}^{m} \sum_{j} (p_{ij})^2 \quad (6)$$

Among them, $b$ is the PCA processed data mentioned earlier, $\hat{b}$ is the data reconstructed by the SAGE model, and $p$ is an iterative variable that represents each trainable parameter in the model. The model continuously updates node features by iterating through *layer*, gradually learning more feature representations. Multiply the adjacency sparse matrix with the node dense matrix to obtain a new feature matrix, process it through the batch normalization layer, perform feature transformation through the linear layer, and then introduce nonlinearity using the LeakyReLU activation function to gradually learn more feature representations, and finally reconstruct the image. The specific model structure is shown in Fig 2.

After obtaining the reconstructed image, calculate the residual between the original image and the reconstructed image. Since the target pixel is relatively small compared to the background pixel, the reconstructed image mainly contains background information, and the residual can highlight the target pixel. Then use the CEM method to process the residual data to achieve target detection. To eliminate the presence of background noise, a nonlinear mapping function [10] is introduced. By suppressing the function to retain a larger output response and suppressing a smaller output response, the background will be suppressed while the target will be preserved in the final detection result. The specific formula is as follows:

$$D_{final}(c) = \begin{cases} 1 - e^{-\alpha c} & , c \geq 0 \\ 0 & , c < 0 \end{cases} \quad (7)$$

In the formula, $c$ is the data obtained using the CEM detector after residual, $\alpha$ is a fixed parameter of $1e{-}4$, and $D_{final}$ is the final detection result. The effect diagram of the detection process for each part of SAGE is shown in Fig 3. Summary of the complete detection scheme for the SAGE target detection model in Algorithm 1.

The overall parameters of the experiment in this article are given in Table 1, which can be used as a reference (after multiple experiments, it was found that changing these parameters had little effect on the detection results). Among them, *hidden–dim* refers to the number of hidden layers in the SAGE model, *layer* refers to the number of convolutional layers in the graph, and *epoch* refers to the number of training iterations in the dataset, $\alpha$ (as mentioned above) is the nonlinear mapping parameter.

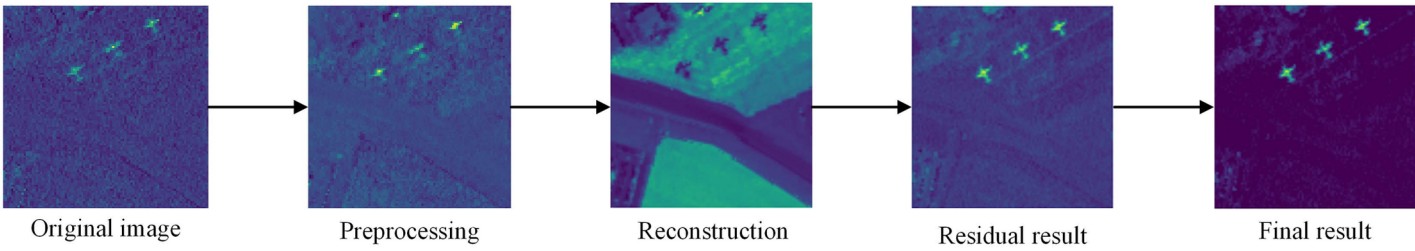

| Original image | Preprocessing | Reconstruction | Residual result | Final result |

**Fig 3. Effect diagram of detection process.**

**Table 1. Parameter settings for the proposed method.**

| Parameter | hidden-dim | layer | epochs | $\alpha$ |
|---|---|---|---|---|
| Value | 128 | 3 | 2000 | $1 \times 10^{-4}$ |

**Algorithm 1.** SAGE for HSI Target Detection

**Input:** HSI-tensor x, prior target spectrum d.
1. Normalize the input data x to obtain $x_n$, and enhance its features using the tanh function, as shown in Eq (3) and Eq (4).
2. Use PCA to transform the dimensionality of the enhanced results to obtain b , and obtain the total number of pixels N.
3. Build graph structure: Traverse each pixel in b, consider its surrounding 8 adjacent pixels, calculate the spectral cosine similarity as shown in Eq (5), save the edge index of the undirected graph, and use it and N to construct the adjacency matrix.
4. Initialize the SAGE model based on the data dimension and set the layer parameter for the hidden layer.
**for** *each epoch* **do**
| Minimize the loss function Eq (6).
**end**
5. Reconstruct the image through the model to obtain $\hat{b}$. Subtract $\hat{b}$ from the input data b to obtain the residual, and use CEM for detection.
6. Input the residual data detected by CEM into a nonlinear mapping function to filter out background noise, as shown in the Eq (7).

## Experimental results and analysis

In order to verify the effectiveness of the method proposed in this paper, real hyperspectral data were selected for experiments. All experiments were conducted on NVIDIA GeForce GTX1650 computers using version 2.1 of the PyTorch framework and version 3.9 of Python.

### Dataset introduction

To comprehensively and effectively evaluate the performance of the method proposed in this paper, 7 representative publicly available HSI datasets were selected for the experiment. The information on these datasets is shown in Table 2. These datasets have been preprocessed during download to remove noise bands such as water vapor that have less impact on noise. This not only reduces computational complexity but also optimizes data quality to a certain extent, making the experimental results more focused on the effectiveness evaluation of target detection algorithms. Due to multiple factors affecting the capture of each hyperspectral dataset, even with the same instrument, the same wavelength band can have different effects on detection. The detailed introduction is as follows:

**Table 2. Information from hyperspectral datasets.**

| Dataset | Space Size (Pixel) | Bands (Piece) | Target Pixels (Each) | Spatial Res. (Meter) |
|---|---|---|---|---|
| Segundo | $250 \times 300$ | 224 | 2048 | 7.1 |
| Abu-airpt-4 | $100 \times 100$ | 191 | 60 | 3.4 |
| Sandiego1 | $100 \times 100$ | 189 | 57 | 3.5 |
| Sandiego2 | $100 \times 100$ | 189 | 134 | 3.5 |
| Abu-urban-1 | $100 \times 100$ | 204 | 67 | 7.1 |
| Abu-urban-2 | $100 \times 100$ | 207 | 155 | 7.1 |
| HYDICE | $80 \times 100$ | 162 | 21 | 1 |

*Segundo dataset [10].*

Taken by the Airborne Visible/Infrared Imaging Spectrometer (AVIRIS), there are 224 available bands with a spatial size of $250 \times 300$ pixels and a spatial resolution of 7.1m. The target pixels in this dataset are the construction area of the refinery, with a total of 2048 pixels. The characteristic of this dataset lies in its relatively large spatial size and large number of bands, which can fully test the algorithm's ability to process large-scale hyperspectral data. At the same time, as the target of the refinery construction area, its spectral characteristics have certain differences from the surrounding background, which puts high demands on the algorithm's target discrimination ability.

*Sandiego dataset.*

Captured by AVIRIS Naval Airport covering the San Diego area. The original dataset had 224 spectral bands, and after removing water vapor absorption and noise bands, 189 bands were retained with a spatial size of $100 \times 100$ pixels and a spatial resolution of approximately 3.5m. This dataset targets three aircraft. Due to the large amount of data, it was cropped into two sub-datasets for easy detection, with targets occupying 57 and 134 pixels respectively. The Sandiego dataset has a large number of spectral bands and moderate spatial resolution. Aircraft targets in complex airport backgrounds have relatively small target features and high spectral confusion with the background, which poses a serious challenge to the algorithm's ability to detect small targets in complex backgrounds.

*ABU dataset [24].*

This dataset contains multiple subsets of data from different scenarios, all with a spatial size of $100 \times 100$ pixels and a spatial resolution of 7.1m (except for the abu-airport-4 subset). The abu-airport-4 dataset was obtained from hyperspectral images of the Gulfport region, with a spatial resolution of 3.4 meters and three aircraft targets. It has a total of 191 bands and 60 target pixels; The abu-urban-1 subset has 204 spectral bands and 67 residential target pixels; The abu-urban-2 subset has 207 spectral bands and 155 residential target pixels. The diversity of the ABU dataset is reflected in different target types (aircraft and residential) and the number of spectral bands, which helps to comprehensively evaluate the adaptability and effectiveness of the algorithm in various target scenarios and data features.

*HYDICE dataset.*

The data of a city area in California, USA, recorded by the Hyperspectral Digital Image Acquisition Experiment (HYDICE) sensor, has an image size of $80 \times 100 \times 162$ pixels, with cars and roofs labeled, totaling 21 pixels. The characteristic of this dataset is a spatial resolution of 1m, which is relatively high and can provide more detailed ground information. However, the number of target pixels is relatively small, which requires high requirements for algorithms

in small target detection and accurate positioning. At the same time, the complex ground environment in urban areas also increases the difficulty of target detection.

## Evaluating indicator

In order to comprehensively and objectively evaluate the performance of the model in hyperspectral target detection, a series of comprehensive evaluation indicators were adopted, among which the ROC [25] (Receiver Operating Characteristic) curve and its related indicators were particularly critical. The ROC curve describes the detection performance of the true positive rate ($P_d$) and false positive rate ($P_f$) for each value $\tau$. The ($P_f$, $P_d$) curve describes the relationship between the detection rate and the false alarm rate. The formulas for $P_d$ and $P_f$ are:

$$P_d = \frac{TP}{TP + FN}, P_f = \frac{FP}{FP + TN} \tag{8}$$

Among them, TP (True Positive), which refers to the number of pixels that are actually targets and correctly detected as targets; FN (False Negative), which refers to the number of pixels that are actually targets but mistakenly detected as backgrounds; FP(False Positive), which refers to the number of pixels that are actually background but mistakenly detected as targets; TN (True Negative), which refers to the number of pixels that are actually the background and correctly detected as the background.

Calculate the area under the ROC curve, denoted as $AUC(P_f, P_d)$, $AUC(\tau, P_d)$, and $AUC(\tau, P_f)$, for quantitative evaluation. For $AUC(P_f, P_d)$ and $AUC(\tau, P_d)$, higher values indicate better results, where 1 is the best value. Conversely, in the case of $AUC(\tau, P_f)$, a lower value is preferred, where 0 is the best value. In addition, we consider two combined criteria [12], incorporating the three AUC values into an overall evaluation as $AUC_{OA}$ and $AUC_{SNPR}$, which are expressed as follows.

$$AUC_{OA} = AUC(P_f, P_d) + AUC(\tau, P_d) - AUC(\tau, P_f) \tag{9}$$

$$AUC_{SNPR} = \frac{AUC(\tau, P_d)}{AUC(\tau, P_f)} \tag{10}$$

In these two criteria, the higher the values of $AUC_{OA}$ and $AUC_{SNPR}$, the better the detection performance. $AUC(P_f, P_d)$ represents the overall effectiveness of detection. $AUC(\tau, P_d)$ represents the effectiveness of target detection. $AUC(\tau, P_f)$ represents the effectiveness of background suppression.

## Experimental result

For the detection of the hyperspectral dataset introduced above, the comparison of detection results using different methods with average spectra is shown in Fig 4, including CEM [4], ACE [5], SSROW [16], BLTSC [10], HTD-IRN [12], and SAGE. The datasets from top to bottom are Segundo abu-airport-4, Sandiego1, Sandiego2, abu-urban-1, abu-urban-2, and HYDICE.

Although machine learning methods CEM and ACE can detect targets in some datasets, there are significant shortcomings. They are not ideal in handling the relationship between targets and backgrounds, with poor separation of targets and backgrounds, and there is a lot of noise in the detection results. This is mainly because its algorithm parameters are relatively fixed, making it difficult to adapt to the complex pixel relationships and changes in target

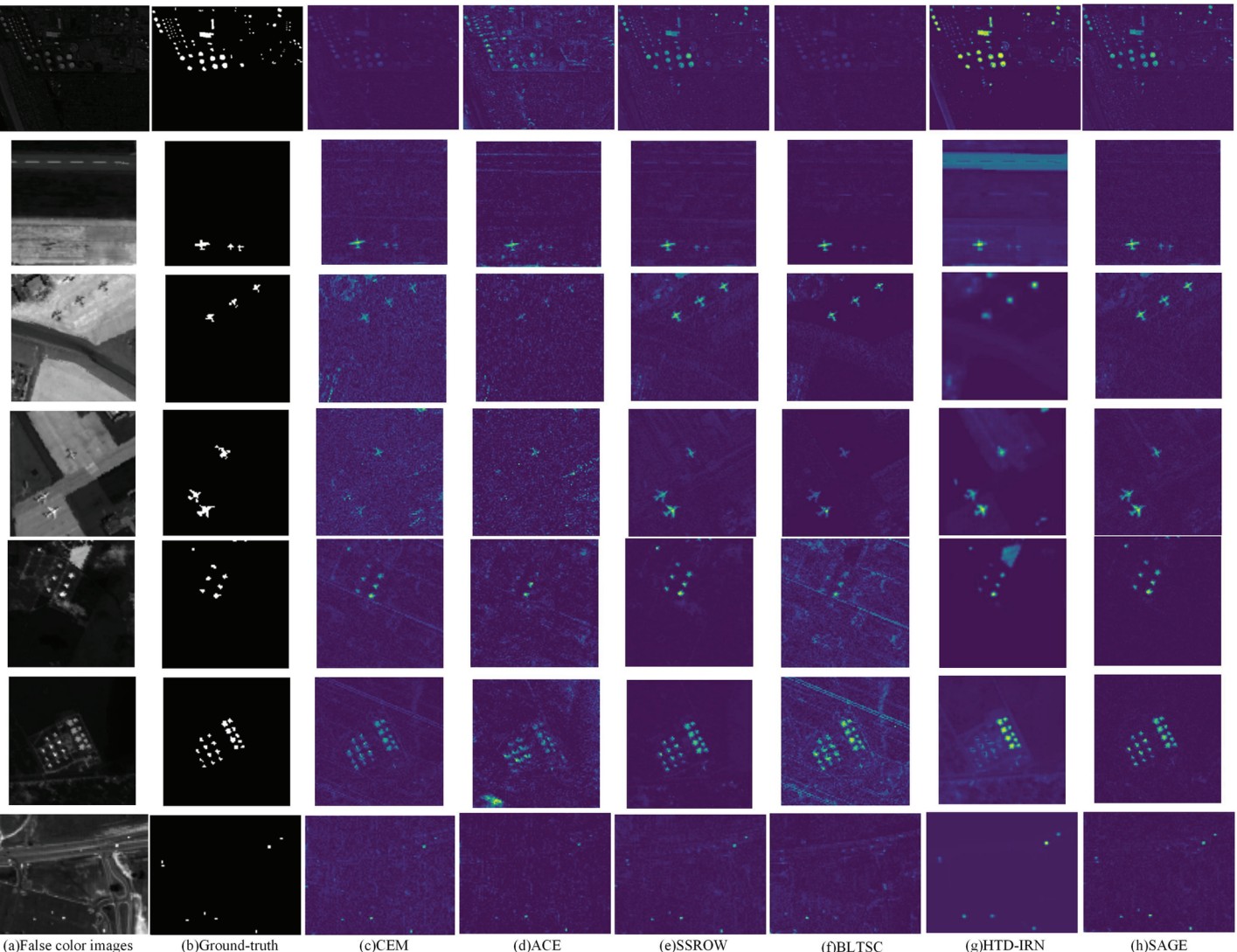

(a)False color images (b)Ground-truth (c)CEM (d)ACE (e)SSROW (f)BLTSC (g)HTD-IRN (h)SAGE

**Fig 4. Detection consequence of diverse methods**.

morphology in hyperspectral data; SSROW can separate targets from background pixels and detect them well, but the process is relatively complex; BLTSC can detect targets well on certain datasets, but it requires a long training time. In the detection effect map, the edge and detail information of the target are not clear enough, indicating that its extraction and representation of target features are not accurate enough; HTD-IRN can detect the target on some datasets, but the target patterns are somewhat blurred from the effect image, making it difficult to accurately identify the detailed features of the target; The SAGE method in this article can separate target pixels from the background, effectively detect targets, and has relatively less noise. At the same time, it has the advantages of short training time and high accuracy, and the process is not complicated.

Although ROC curves can provide insights into detection performance, it is often difficult to accurately evaluate results due to overlapping results from multiple methods. Therefore, we compared in detail the five AUC metrics obtained from 7 datasets in Tables 3, 4, 5, 6, 7, 8,

**Table 3. Quantitative detection results obtained using all methods on the *Segundo* dataset. The best and second best are marked in bold and underlined respectively.**

| Method | $\mathrm{AUC}(\mathbf{P_f, P_d})$ | $\mathrm{AUC}(\tau, \mathbf{P_f})$ | $\mathrm{AUC}(\tau, \mathbf{P_d})$ | $\mathrm{AUC_{OA}}$ | $\mathrm{AUC_{SNPR}}$ |
|---|---|---|---|---|---|
| CEM | 0.9354 | 0.7381 | <u>0.8020</u> | 0.9993 | 1.0866 |
| ACE | 0.6273 | 0.0515 | 0.1225 | 0.6980 | 2.3774 |
| SSROW | <u>0.9971</u> | 0.1828 | 0.5090 | 1.3239 | 2.7878 |
| BLTSC | 0.9358 | 0.8137 | **0.8742** | 0.9964 | 1.0743 |
| HTD-IRN | 0.9939 | <u>0.0134</u> | 0.6221 | **1.6020** | **46.3668** |
| SAGE | **0.9985** | **0.0113** | 0.5249 | <u>1.4022</u> | <u>36.8691</u> |

**Table 4. Quantitative detection results obtained using all methods on the *abu–airport–4* dataset. The best and second best are marked in bold and underlined respectively.**

| Method | $\mathrm{AUC}(\mathbf{P_f, P_d})$ | $\mathrm{AUC}(\tau, \mathbf{P_f})$ | $\mathrm{AUC}(\tau, \mathbf{P_d})$ | $\mathrm{AUC_{OA}}$ | $\mathrm{AUC_{SNPR}}$ |
|---|---|---|---|---|---|
| CEM | 0.8702 | 0.2178 | 0.5448 | 1.1972 | 2.5011 |
| ACE | 0.8363 | <u>0.0265</u> | 0.3522 | 1.1620 | <u>13.2790</u> |
| SSROW | 0.9981 | 0.1448 | <u>0.5542</u> | <u>1.4074</u> | 3.8258 |
| BLTSC | <u>0.9982</u> | 0.1123 | 0.4903 | 1.3974 | 5.3650 |
| HTD-IRN | 0.9398 | 0.0951 | 0.4579 | 1.2878 | 4.6578 |
| SAGE | **0.9994** | **0.0131** | **0.6082** | **1.5182** | **40.6695** |

**Table 5. Quantitative detection results obtained using all methods on the *Sandiego*1 dataset. The best and second best are marked in bold and underlined respectively.**

| Method | $\mathrm{AUC}(\mathbf{P_f, P_d})$ | $\mathrm{AUC}(\tau, \mathbf{P_f})$ | $\mathrm{AUC}(\tau, \mathbf{P_d})$ | $\mathrm{AUC_{OA}}$ | $\mathrm{AUC_{SNPR}}$ |
|---|---|---|---|---|---|
| CEM | 0.9512 | 0.5854 | **0.7920** | 1.1578 | 1.3528 |
| ACE | 0.8972 | 0.0326 | 0.2067 | 1.0714 | 6.3422 |
| SSROW | <u>0.9976</u> | 0.2234 | <u>0.7244</u> | <u>1.4880</u> | 1.1953 |
| BLTSC | 0.9921 | 0.2907 | 0.6388 | 1.3462 | 2.1901 |
| HTD-IRN | 0.9975 | **0.0023** | 0.4579 | 1.4531 | **201.7444** |
| SAGE | **0.9990** | <u>0.0191</u> | 1.6441 | **1.6241** | <u>33.7808</u> |

and 9). The data in the tables were all normalized and detected, with bold values indicating the optimal values for that row and underlined values indicating suboptimal values. Analysis of these table data shows that SAGE has achieved excellent results on each dataset, with an average improvement of approximately 9.6% and 22% compared to traditional CEM and ACE detection methods, respectively. It is also 0.5% and 3.3% higher than SSROW and BLTSC models based on autoencoders and variational encoders, and 1.7% higher than HTD-IRN models based on physical interpretability. On some datasets, even if the SAGE model does not achieve optimal effectiveness in target detection and background suppression, the difference is not significant. The SAGE model demonstrates strong competitiveness in the field of hyperspectral target detection, with high comprehensive evaluation scores on all 7 datasets. In this field, targets have rare spectral features, and SAGE models learn based on inter-node relationships, which can more effectively capture these uncommon pixels.

The three-dimensional ROC curve, i.e., the relationship between $P_f$, $P_d$, and $\tau$, is shown in Fig 5. From the figure, it can be seen that the SAGE method shows better detection performance in several hyperspectral datasets and is able to obtain a higher true positive rate with a lower false positive rate, and the performance is more stable. For example, on the Segundo dataset, the ROC curve of the SAGE model is significantly better than that of other methods. In most of the datasets, the ROC curves of the traditional CEM and ACE methods have lower true alarm rate (TAR) values at low false alarm rate (FAR), indicating that these two methods

**Table 6. Quantitative detection results obtained using all methods on the *Sandiego*2 dataset. The best and second best are marked in bold and underlined respectively.**

| Method | AUC($P_f, P_d$) | AUC($\tau, P_f$) | AUC($\tau, P_d$) | AUC$_{OA}$ | AUC$_{SNPR}$ |
|---|---|---|---|---|---|
| CEM | 0.8049 | 0.5262 | **0.6256** | 0.9043 | 1.1889 |
| ACE | 0.6918 | 0.0482 | 0.1399 | 0.7834 | 2.8968 |
| SSROW | <u>0.9976</u> | 0.1111 | <u>0.5209</u> | <u>1.4073</u> | 4.6881 |
| BLTSC | 0.9715 | 0.1505 | 0.3704 | 1.1915 | 2.4609 |
| HTD-IRN | 0.9907 | <u>0.0187</u> | 0.4040 | 1.3759 | <u>21.5245</u> |
| SAGE | **0.9980** | **0.0129** | 0.4434 | **1.4285** | **34.1746** |

**Table 7. Quantitative detection results obtained using all methods on the *abu–urban–*1 dataset. The best and second best are marked in bold and underlined respectively.**

| Method | AUC($P_f, P_d$) | AUC($\tau, P_f$) | AUC($\tau, P_d$) | AUC$_{OA}$ | AUC$_{SNPR}$ |
|---|---|---|---|---|---|
| CEM | 0.7944 | 0.3472 | 0.5179 | 0.9651 | 1.4918 |
| ACE | 0.7232 | 0.0451 | 0.1297 | 0.8078 | 2.8771 |
| SSROW | <u>0.9993</u> | 0.0579 | <u>0.5537</u> | <u>1.4951</u> | 9.5605 |
| BLTSC | 0.9738 | 0.4111 | **0.6219** | 1.1847 | 1.5129 |
| HTD-IRN | 0.9869 | <u>0.0074</u> | 0.2947 | 1.2740 | <u>39.7981</u> |
| SAGE | **0.9996** | **0.0071** | 0.5077 | **1.5191** | **71.5528** |

**Table 8. Quantitative detection results obtained using all methods on the *abu–urban–*2 dataset. The best and second best are marked in bold and underlined respectively.**

| Method | AUC($P_f, P_d$) | AUC($\tau, P_f$) | AUC($\tau, P_d$) | AUC$_{OA}$ | AUC$_{SNPR}$ |
|---|---|---|---|---|---|
| CEM | 0.9867 | 0.2257 | 0.5254 | 1.2865 | 2.3278 |
| ACE | 0.7308 | <u>0.0475</u> | 0.1114 | 0.8422 | 2.6928 |
| SSROW | **0.9991** | 0.0665 | 0.5739 | **1.5064** | 8.6249 |
| BLTSC | 0.9456 | 0.5009 | **0.7494** | 1.1940 | 1.4959 |
| HTD-IRN | 0.9621 | 0.0536 | 0.5085 | <u>1.4170</u> | <u>9.4734</u> |
| SAGE | <u>0.9981</u> | **0.0091** | <u>0.6176</u> | 1.3714 | **41.7888** |

**Table 9. Quantitative detection results obtained using all methods on the *HYDICE* dataset. The best and second best are marked in bold and underlined respectively.**

| Method | AUC($P_f, P_d$) | AUC($\tau, P_f$) | AUC($\tau, P_d$) | AUC$_{OA}$ | AUC$_{SNPR}$ |
|---|---|---|---|---|---|
| CEM | 0.9754 | 0.1895 | 0.6097 | 1.3956 | 3.2165 |
| ACE | 0.9442 | **0.0407** | 0.3510 | <u>1.4731</u> | 8.6198 |
| SSROW | <u>0.9998</u> | 0.1560 | <u>0.6274</u> | 1.4712 | 4.0955 |
| BLTSC | 0.9407 | 0.1777 | 0.4198 | 1.2187 | 2.5696 |
| HTD-IRN | 0.9975 | <u>0.0414</u> | 0.4698 | 1.4258 | **11.3388** |
| SAGE | **0.9999** | 0.0567 | **0.6296** | **1.5728** | <u>11.0939</u> |

are prone to produce more False positive results during detection. For the other deep learning methods (SSROW, BLTSC, and HTD-IRN), these methods show some advantages on individual datasets but are not as good as SAGE on other datasets. e.g., HTD-IRN performs comparably to SAGE on the Sandiego1 dataset but performs relatively poorly on the abu-airport-4 dataset.

Fig 6 box plot shows the performance of various methods in separating targets and backgrounds on seven datasets. The larger the distance and the smaller the overlap between targets and backgrounds, the better the separation degree. SAGE has shown excellent separation performance on most datasets, for example, on the Sandiego1 dataset, compared with other

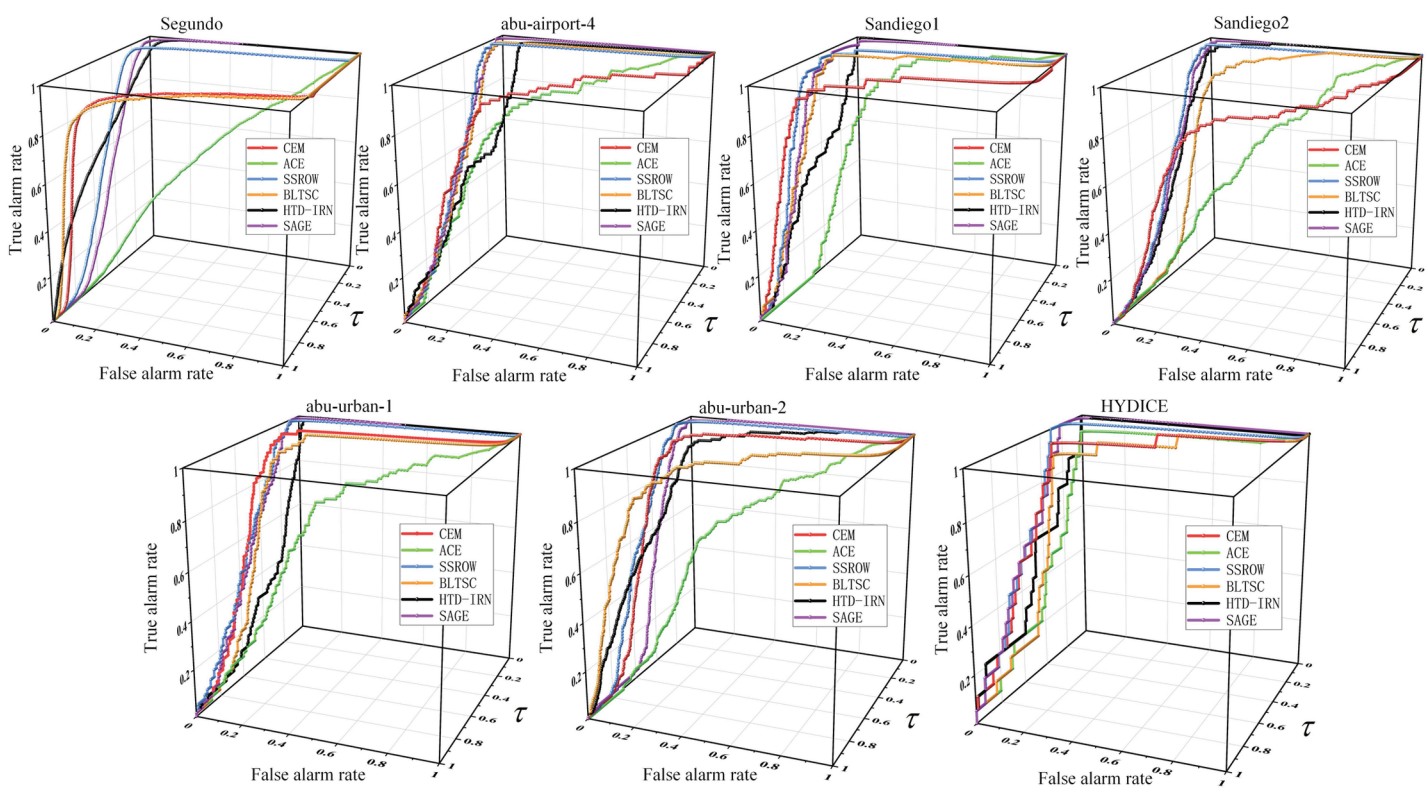

**Fig 5. Comparison of 3D ROC curves under different methods.**

methods, the SAGE method shows the maximum distance and minimum overlap between the target and background.

This means that the SAGE model can more effectively separate targets from complex backgrounds, improving the accuracy of target detection. Although HTD-IRN performs well in suppressing background, their overall performance in accuracy and target background separation is not as good as SAGE models. Other methods have varying degrees of shortcomings in suppressing background noise, target detection, and target background separation, further highlighting the advantages of SAGE models in handling hyperspectral target detection tasks.

Table 10 lists the running time of the proposed and compared methods on different datasets, including training time and running time. Compared with two traditional machine learning methods (CEM and ACE), the SAGE method in this article takes more time to train the model, but this is due to the complexity of deep learning models and the deep requirements for data feature learning. Compared with the other three deep learning models (SSROW, BLTSC, and HTD-IRN), the SAGE model has a certain advantage in terms of time, and even takes much less time than the two of them. This indicates that the SAGE model can complete tasks in a relatively short time while ensuring detection accuracy, improving the efficiency of the algorithm.

In order to analyze the robustness of the SAGE method and comparative method under noise conditions, 10dB Gaussian noise was added to the original image. From Table 11, it can be seen that the SAGE model still maintains good detection performance on most datasets, and the average accuracy of the SAGE model is much higher than that of the BLTSC and HTD-IRN models. Compared to the absence of noise, the AUC value of the method proposed

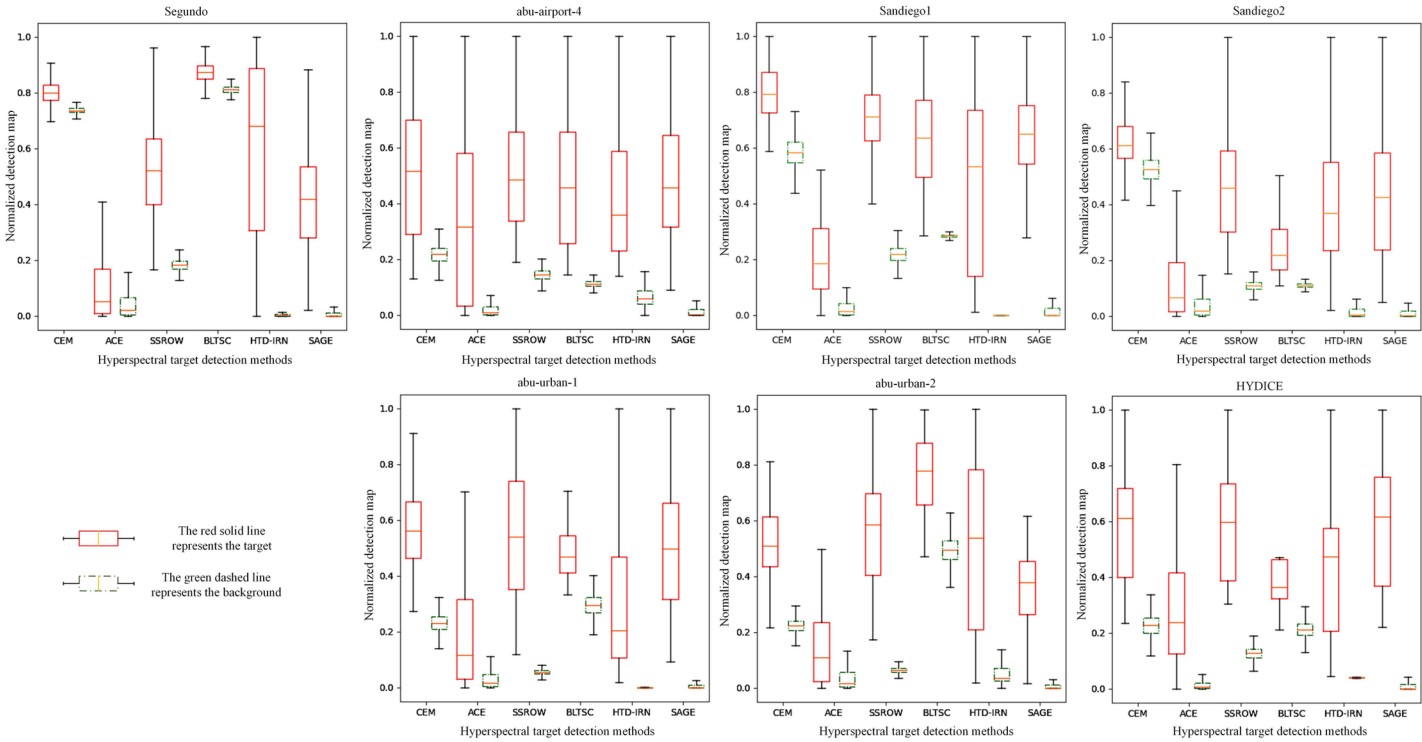

**Fig 6. Comparison of target and background separation degree among various methods**.

**Table 10. Total running time of various methods on different datasets (unit: s)**

| Dataset | CEM | ACE | SSROW | BLTSC | HTD-IRN | SAGE |
|---|---|---|---|---|---|---|
| Segundo | 0.33 | 12.20 | 488.61 | 1194.51 | 1711.17 | 715.62 |
| abu-airport-4 | 0.03 | 1.64 | 87.65 | 148.04 | 214.01 | 93.69 |
| Sandiego1 | 0.03 | 1.63 | 93.97 | 149.33 | 209.10 | 94.07 |
| Sandiego2 | 0.08 | 1.61 | 96.22 | 150.45 | 217.22 | 79.73 |
| abu-urban-1 | 0.03 | 1.69 | 58.33 | 147.68 | 231.43 | 100.20 |
| abu-urban-2 | 0.03 | 1.65 | 59.38 | 147.68 | 241.05 | 103.50 |
| HYDICE | 0.02 | 1.29 | 68.88 | 102.47 | 220.26 | 73.29 |

**Table 11. AUC values of various methods under 10$dB$ Gaussian noise. The best values are marked in bold.**

| Dataset | CEM | ACE | SSROW | BLTSC | HTD-IRN | SAGE |
|---|---|---|---|---|---|---|
| Segundo | 0.7378 | 0.7585 | 0.9118 | 0.9365 | **0.9937** | 0.9911 |
| abu-airport-4 | 0.8154 | 0.8441 | 0.9895 | **0.9978** | 0.8906 | 0.9835 |
| Sandiego1 | 0.9501 | 0.9103 | 0.9827 | 0.9910 | 0.9966 | **0.9986** |
| Sandiego2 | 0.8124 | 0.7115 | 0.9371 | 0.9637 | **0.9929** | 0.9923 |
| abu-urban-1 | 0.9779 | 0.8754 | 0.9938 | 0.9656 | 0.9841 | **0.9976** |
| abu-urban-2 | 0.8282 | 0.7991 | 0.9983 | 0.9440 | 0.9886 | **0.9989** |
| HYDICE | 0.9811 | 0.9851 | 0.9959 | 0.9449 | 0.9409 | **0.9979** |
| Average | 0.8718 | 0.8405 | 0.9727 | 0.9633 | 0.9696 | **0.9942** |

in this paper has decreased, mainly due to the interference of noise between the prior target spectrum and the target to be detected after adding noise, resulting in a decrease in detection accuracy.

## Conclusion

The hyperspectral target detection model SAGE based on graph sampling and aggregation network proposed in this article automatically learns effective feature representations of nodes in the graph, achieves feature extraction and processing of graph data, and can fully utilize pixel relationships for feature learning. It also exhibits good robustness in complex backgrounds and noisy environments, outperforming other comparative methods. Based on the similarity between 8-neighborhood pixels and spectral cosine, a graph structure is constructed. By selecting edges with higher similarity through a threshold, the relationships between pixels are reflected through nodes and edges in the graph structure, effectively capturing irregular structures between pixels and extracting more representative features. Through extensive and in-depth experiments on seven hyperspectral image datasets, the SAGE model has demonstrated outstanding performance. Its average detection accuracy is over 99.8%, which fully demonstrates the high-precision ability of the model in hyperspectral target detection. At the same time, the SAGE model has shown excellent adaptability when dealing with different datasets. Whether it is the spatial resolution of the dataset, the difference in the number of spectral bands, or the diversity of target types and distributions, the SAGE model can flexibly respond and always maintain high detection accuracy. It also performs well in terms of time efficiency, spending less time compared to other methods and achieving efficient and accurate target detection.

Although the SAGE model has achieved good results in the field of hyperspectral target detection, there is still room for improvement. The current model has a small amount of redundant noise in the background during detection, which is a key direction for future improvement. The follow-up research plan is to further optimize the algorithm for constructing graph structures, explore more effective feature extraction and aggregation strategies, and deeply mine pixel relationships, in order to minimize the impact of background noise on detection results and continuously improve model performance.

## Supporting information

**S1 Fig. Details of Fig 5.** Fig 5 compares the 3D ROC curves of each dataset using different methods The .xlsx file stores the experimental data of the hyperspectral dataset. Each dataset corresponds to a sub-table, with rows representing the three parameters corresponding to each method. The processed data shows performance differences in detection rate and threshold methods.
(XLSX)

**S2 Fig. Details of Fig 6.** Fig 6 compares the degree of target background separation of various methods The .xlsx file records the data for each method's target and background, with each sub-table corresponding to a dataset. The analysis of data highlights the advantages of the SAGE model in separation.
(XLSX)

## Author contributions

**Conceptualization:** Hongfeng Jin, Zhiqiu Li.

**Data curation:** Hongfeng Jin.

**Formal analysis:** Hongfeng Jin, Zhiqiu Li.

**Funding acquisition:** Tie Li.

**Investigation:** Hongfeng Jin.

**Project administration:** Tie Li, Hongfeng Jin.

**Resources:** Hongfeng Jin.

**Software:** Hongfeng Jin.

**Supervision:** Tie Li.

**Validation:** Tie Li, Hongfeng Jin.

**Visualization:** Hongfeng Jin.

**Writing – original draft:** Hongfeng Jin.

**Writing – review & editing:** Tie Li.

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
