## [Decision Letter · Decision Letter 0]

8 Dec 2024

PONE-D-24-42135Hyperspectral target detection based on graph sampling and aggregation networkPLOS ONE

Dear Dr. jin,

Thank you for submitting your manuscript to PLOS ONE. After careful consideration, we feel that it has merit but does not fully meet PLOS ONE’s publication criteria as it currently stands. Therefore, we invite you to submit a revised version of the manuscript that addresses the points raised during the review process.

We look forward to receiving your revised manuscript.

Kind regards,

Panos Liatsis, PhD

Academic Editor

PLOS ONE

Journal Requirements:

2. Please note that PLOS ONE has spec6ific guidelines on code sharing for submissions in which author-generated code underpins the findings in the manuscript. In these cases, all author-generated code must be made available without restrictions upon publication of the work. Please review our guidelines at https://journals.plos.org/plosone/s/materials-and-software-sharing#loc-sharing-code and ensure that your code is shared in a way that follows best practice and facilitates reproducibility and reuse.

4. We note that your Data Availability Statement is currently as follows: “All relevant data are within the manuscript and in Supporting Information files.”

Please confirm at this time whether or not your submission contains all raw data required to replicate the results of your study. Authors must share the “minimal data set” for their submission. PLOS defines the minimal data set to consist of the data required to replicate all study findings reported in the article, as well as related metadata and methods (https://journals.plos.org/plosone/s/data-availability#loc-minimal-data-set-definition). For example, authors should submit the following data: - The values behind the means, standard deviations and other measures reported; - The values used to build graphs; - The points extracted from images for analysis. Authors do not need to submit their entire data set if only a portion of the data was used in the reported study. If your submission does not contain these data, please either upload them as Supporting Information files or deposit them to a stable, public repository and provide us with the relevant URLs, DOIs, or accession numbers. For a list of recommended repositories, please see https://journals.plos.org/plosone/s/recommended-repositories. If there are ethical or legal restrictions on sharing a de-identified data set, please explain them in detail (e.g., data contain potentially sensitive information, data are owned by a third-party organization, etc.) and who has imposed them (e.g., an ethics committee). Please also provide contact information for a data access committee, ethics committee, or other institutional body to which data requests may be sent. If data are owned by a third party, please indicate how others may request data access.

5. Please ensure that you refer to Figure 5 and 6 in your text as, if accepted, production will need this reference to link the reader to the figure.

6. We note you have included a table to which you do not refer in the text of your manuscript. Please ensure that you refer to Table 1-12 in your text; if accepted, production will need this reference to link the reader to the Table.

Additional Editor Comments:

Improve the analysis of the state-of-the-art by explaining the advantages and disadvantages of previous work so that you can better position the definition of the contributions of this work.Expand the explanation of the datasets and scientific methodology to improve understanding and repeatabilityProvide a comparison in terms of noise robustness with the state-of-the-artImprove the explanation of the impact of the work in the conclusions

Reviewers' comments:

Reviewer's Responses to Questions

**Comments to the Author**

1. Is the manuscript technically sound, and do the data support the conclusions?

Reviewer #1: Yes

Reviewer #2: Yes

2. Has the statistical analysis been performed appropriately and rigorously? 

Reviewer #1: Yes

Reviewer #2: Yes

3. Have the authors made all data underlying the findings in their manuscript fully available?

Reviewer #1: Yes

Reviewer #2: Yes

4. Is the manuscript presented in an intelligible fashion and written in standard English?

Reviewer #1: Yes

Reviewer #2: Yes

5. Review Comments to the Author

Reviewer #1: The paper proposed a graph sampling aggregation network model for hyperspectral object detection. Graph sampling was used to reduce computational complexity, and aggregation was used to combine neighbor information and update node features. The model had been tested on multiple hyperspectral image datasets and compared with other models. The experiment is sufficient, and the result analysis is comprehensive and described in detail.

There are the following problems need further modification and improvement:

1. In Section I. INTRODUCTION, after reviewing the research status of using deep learning to construct hyperspectral object detection models, the existing problems in current research should be summarized, and a brief description should be provided on how these problems would be solved with the new proposed model.

2. It is suggested to add a schematic diagram in part B of Section II, and conduct a more sufficient and clear description of building graph structure with the diagram.

3. Section IV of the paper should be the conclusion rather than the acknowledgement. Although the conclusion was supported by the experimental results, it did not clearly indicate the advantages of the model when dealing with irregular structures in hyperspectral data, which had been specified in Section I.

Reviewer #2: The authors propose a hyperspectral target detection model, SAGE, based on graph sampling and aggregation networks, designed to better utilize the spectral information of complex hyperspectral images and handle the irregular structures between pixels.

The overall framework aims to address the challenges in hyperspectral target detection tasks, such as high computational complexity, limited feature learning efficiency, and the difficulty in capturing both local and global dependencies simultaneously. To reduce computational complexity, the authors introduce a graph sampling mechanism that selects a subset of neighboring nodes for feature learning and propagation, enabling effective learning of node feature representations within the graph and improving efficiency. To ensure detection accuracy and effective graph data processing, the authors use aggregation networks and integrate graph convolution networks into the SAGE model, allowing information to propagate across multiple steps while capturing both local and global dependencies. Experimental results on multiple datasets show that the proposed method demonstrates certain superiority.

I carefully reviewed the manuscript and have five concerns:

Please see my comments for details. I hope that the authors could reap benefits from the comments.

Comments:

(1) In the related work section, although the characteristics of traditional methods (such as CEM and ACE) and deep learning methods (such as HTD-IRN) are listed, the innovative aspects of the SAGE method and its comparison with existing methods are not fully elaborated.

(2) In Table II, the setting of the regularization parameter λ and the cosine similarity threshold ρ is not explained in detail. Have you considered introducing an adaptive strategy to dynamically adjust the graph structure or parameters?

(3) Could you provide a deeper discussion on how the model adapts to different datasets? Is the model fine-tuned for each dataset individually, or is a unified setting used across all datasets?

(4) In the noise disturbance experiments, the lack of comparison with other methods may not fully highlight the advantages of SAGE in this regard. Additionally, have the authors considered incorporating more evaluation metrics, rather than solely relying on AUC?

(5) The article overall seems not too bad, and I recommend a thorough re-review to improve the clarity, accuracy, and overall coherence of the paper.

6. PLOS authors have the option to publish the peer review history of their article (what does this mean?). If published, this will include your full peer review and any attached files.

Reviewer #1: No

Reviewer #2: No

---

## [Author Response · Author response to Decision Letter 1]

23 Dec 2024

Response to Reviewers

Dear academic editor and reviewer, hello! I carefully reviewed my manuscript and found that the explanation was unclear and the description was insufficient. I have made the corresponding modifications, please refer to the attachment if possible. The response to each viewpoint presented in the manuscript is as follows:

Editor Comments:

The opinions are as follows:

1.Improve the analysis of the state-of-the-art by explaining the advantages and disadvantages of previous work so that you can better position the definition of the contributions of this work.

Answer: I have reorganized the advantages of the methods mentioned in Chapter 1 and added the disadvantages of these methods. Specifically, as follows:

Detectors based on Constrained Energy Minimization (CEM) may overlook the influence of target pixels when estimating background pixels; Detectors based on adaptive consistency/cosine estimator (ACE) typically assume that there is additional noise in the background and not in the target, which often does not hold true in practical situations. In order to delve deeper into the intrinsic properties of HSI data, many researchers have been working to develop more complex modeling techniques, such as the E-CEM detection method, which requires multiple samples with feedback and the construction of multiple detectors. When the parameters are small, the algorithm performance is not stable enough; The kernel-based OSP detector maps the original space to the kernel space, enabling traditional OSP [8] methods to effectively utilize the nonlinear characteristics of HSI. However, this mapping process may result in the loss or distortion of some data information.

Xie et al. proposed a deep learning algorithm BLTSC based on background learning, which requires inputting background samples into the AAE network model for training. However, the AAE network structure is complex and requires a large amount of computational resources to learn the feature representation of the background samples during the training process. Meanwhile, BLTSC mainly focuses on background learning and lacks targeted mechanisms for target feature extraction, resulting in poor detection performance on certain datasets.

Shen et al. proposed a detection method HTD-IRN based on interpretable representation networks. Although the physical model is converted into a deep learning network to achieve compatibility between nonlinear representation and physical interpretability, there are still shortcomings in target feature extraction in complex environments, making it difficult to accurately distinguish the boundaries between the target and background when reconstructing them, resulting in detection results only on some datasets.

Zhou et al. proposed a detection algorithm CEM-VAE based on constraint energy minimization variational autoencoder, which uses background to calculate the autocorrelation matrix and introduces CEM regularization term to preserve only background information in the reconstructed samples. However, in the case of small targets or unclear target features, it cannot effectively extract target features, resulting in poor detection performance.

Tian et al. introduced the OSP concept into VAE networks and proposed an orthogonal subspace-guided variational autoencoder learning method for real background representation, OS-VAE, which can train more accurate and reliable background representation models. However, due to its emphasis on background suppression and representation, this method may to some extent neglect the learning of target features.

Li et al. proposed a hyperspectral object detection method SSROW based on spatial-spectral reconstruction and operator weighting. Although it can achieve high-accuracy detection without coarse separation of target and background samples, it requires principal component analysis (PCA) to extract feature vectors, multiple image processing, and operator construction, which increases computational complexity and operational difficulty.

In response to the above issues, the proposed hyperspectral object detection model SAGE based on graph sampling and aggregation networks has significant advantages. In terms of automatic learning feature representation, the SAGE model directly inputs all data into the model and describes the relationships between pixels by constructing adjacency matrices. In graph convolution (GNN) operation, the adjacency sparse matrix is multiplied with the node dense matrix, so that each node can fully obtain the information of its neighboring nodes. Through multiple iterations of updating node features, more representative feature representations are gradually learned. This approach differs from traditional methods and other deep learning models in that it can automatically learn complex relationships between pixels directly from image data, without the need for complex sample preprocessing, and can more accurately capture feature information of small and inconspicuous targets. Compared with other models, the SAGE model can effectively avoid the interference of irregular structures on target detection and accurately identify target pixels when processing hyperspectral data containing complex terrain and targets. In terms of improving detection accuracy and efficiency, the SAGE model reduces unnecessary computational complexity and avoids the problem of duplicate parameter settings by optimizing graph structure construction and graph convolution operations.

2.Expand the explanation of the datasets and scientific methodology to improve understanding and repeatability.

Answer: I have revised the analysis in the experimental results section and added examples after each analysis to improve understanding and reproducibility.

A. In terms of the dataset. I have reorganized the explanation of using the dataset, including spatial size, spatial resolution, and the number of pixels that need to be detected. I also separately added the characteristics of each dataset and the challenges it poses to the model. Specifically, as follows:

To comprehensively and effectively evaluate the performance of the method proposed in this paper, seven representative publicly available HSI datasets were selected for the experiment. The information on these datasets is shown in Table 2. These datasets have been preprocessed during download to remove noise bands such as water vapor that have less impact on noise. This not only reduces computational complexity but also optimizes data quality to a certain extent, making the experimental results more focused on the effectiveness evaluation of object detection algorithms. Due to multiple factors affecting the capture of each hyperspectral dataset, even with the same instrument, the same wavelength band can have different effects on detection. The detailed introduction is as follows:

1) Segundo dataset: captured by the Airborne Visible/Infrared Imaging Spectrometer (AVIRIS), with 224 available bands, a spatial size of 250 × 300 pixels, and a spatial resolution of 7.1m. The target pixels in this dataset are the construction area of the refinery, with a total of 2048 pixels. The characteristic of this dataset lies in its relatively large spatial size and large number of bands, which can fully test the algorithm's ability to process large-scale hyperspectral data. At the same time, as the target of the refinery construction area, its spectral characteristics have certain differences from the surrounding background, which puts high demands on the algorithm's target discrimination ability.

2) Sandiego dataset: captured by AVIRIS Naval Airport covering the San Diego area. The original dataset had 224 spectral bands, and after removing water vapor absorption and noise bands, 189 bands were retained with a spatial size of 100 × 100 pixels and a spatial resolution of approximately 3.5m. This dataset targets three aircraft. Due to the large amount of data, it was cropped into two sub-datasets for easy detection, with targets occupying 57 and 134 pixels respectively. The Sandiego dataset has a large number of spectral bands and moderate spatial resolution. Aircraft targets in complex airport backgrounds have relatively small target features and high spectral confusion with the background, which poses a serious challenge to the algorithm's ability to detect small targets in complex backgrounds.

3) ABU dataset: This dataset contains multiple subsets of data from different scenarios, all with a spatial size of 100 × 100 pixels and a spatial resolution of 7.1m (except for the abu-airport-4 subset). The abu-airport-4 dataset was obtained from hyperspectral images of the Gulfport region, with a spatial resolution of 3.4 meters and three aircraft targets. It has a total of 191 bands and 60 target pixels; The abu-urban-1 subset has 204 spectral bands and 67 residential target pixels; The abu-urban-2 subset has 207 spectral bands and 155 residential target pixels. The diversity of the ABU dataset is reflected in different target types (aircraft and residential) and the number of spectral bands, which helps to comprehensively evaluate the adaptability and effectiveness of the algorithm in various target scenarios and data features.

4) HYDICE dataset: Data from a city area in California, USA, recorded by the Hyperspectral Digital Image Acquisition Experiment (HYDICE) sensor. The image size is 80 × 100 × 162 pixels, with cars and roofs labeled, totaling 21 pixels. The characteristic of this dataset is a spatial resolution of 1m, which is relatively high and can provide more detailed ground information. However, the number of target pixels is relatively small, which requires high requirements for algorithms in small target detection and accurate positioning. At the same time, the complex ground environment in urban areas also increases the difficulty of target detection.

Figure 1 Comparison of 3D-ROC curves under various methods

B．Provide a detailed explanation of the experiment. I reorganized all the experimental analyses in Chapter 3 to make their descriptions more logical and comprehensive. I have changed the original 2D image in Figure 6 to a 3D image, which can better describe the superiority of the experimental method. Meanwhile, I modified Table 11 to include 10dB Gaussian noise in the original image data and compared the detection performance of various methods. Please refer to your opinion 3 for detailed information. Specifically, as follows:

Figure 6 shows the three-dimensional ROC curve, which represents the relationship between Pd, Pf, andτ. From the graph, it can be seen that the SAGE method has shown good detection performance in multiple hyperspectral datasets, achieving high true positive rates with low false positive rates and stable performance. For example, on the Segundo dataset, the ROC curve of the SAGE model is significantly better than that of other methods. In most datasets, the ROC curves of traditional CEM and ACE methods have lower True alarm rate values at low False alarm rates, indicating that these two methods are prone to producing more false positive results during detection. For other deep learning methods (SSROW, BLTSC, and HTD-IRN), these methods have shown certain advantages on individual datasets, but are not as effective as SAGE on other datasets. For example, on the Sandiego1 dataset, HTD-IRN performs similarly to SAGE, but its performance is relatively poor on the abu-airport-4 dataset.

3.Provide a comparison in terms of noise robustness with the state-of-the-art.

Answer：I have modified Table 11 in the original manuscript. Due to the addition of excessive noise, the original image cannot be seen clearly. Therefore, only 10dB Gaussian noise is added to the original image data to compare the detection performance of various methods. The specific analysis is as follows:

In order to analyze the robustness of the SAGE method and comparative method under noise conditions, 10dB Gaussian noise was added to the original image. From Table 11, it can be seen that the SAGE model still maintains good detection performance on most datasets, and the average accuracy of the SAGE model is much higher than that of the BLTSC and HTD-IRN models. Compared to the absence of noise, the AUC value of the method proposed in this paper has decreased, mainly due to the interference of noise between the prior target spectrum and the target to be detected after adding noise, resulting in a decrease in detection accuracy.

Table 1 AUC values of various methods under 10dB Gaussian noise.

The best values are marked in bold.

Dataset CEM ACE SSROW BLTSC HTD-IRN SAGE

Segundo 0.7378 0.7585 0.9118 0.9365 0.9937 0.9911

Abu-airport-4 0.8154 0.8441 0.9895 0.9978 0.8906 0.9835

Sandiego1 0.9501 0.9103 0.9827 0.9910 0.9966 0.9986

Sandiego2 0.8124 0.7115 0.9371 0.9637 0.9929 0.9923

abu-urban-1 0.9779 0.8754 0.9938 0.9656 0.9841 0.9976

abu-urban-2 0.8282 0.7991 0.9983 0.9440 0.9886 0.9989

HYDICE 0.9811 0.9851 0.9959 0.9449 0.9409 0.9979

Average 0.8718 0.8405 0.9727 0.9633 0.9696 0.9942

4.Improve the explanation of the impact of the work in the conclusions

Answer: For the conclusion section, I mainly improved the model from several aspects, including innovation and performance improvement, advantages in handling irregular structures, time efficiency and adaptability, as well as research limitations and prospects. Specifically, as follows:

The hyperspectral object detection model SAGE based on graph sampling and aggregation network proposed in this article automatically learns effective feature representations of nodes in the graph, achieves feature extraction and processing of graph data, and can fully utilize pixel relationships for feature learning. It also exhibits good robustness in complex backgrounds and noisy environments, outperforming other comparative methods. Based on the similarity between 8-neighborhood pixels and spectral cosine, a graph structure is constructed. By selecting edges with higher similarity through a threshold, the relationships between pixels are reflected through nodes and edges in the graph structure, effectively capturing irregular structures between pixels and extracting more representative features. Through extensive and in-depth experiments on seven hyperspectral image datasets, the SAGE model has demonstrated outstanding performance. Its average detection accuracy is over 99.8%, which fully demonstrates the high-precision ability of the model in hyperspectral target detection. At the same time, the SAGE model has shown excellent adaptability when dealing with different datasets. Whether it is the spatial resolution of the dataset, the difference in the number of spectral bands, or the diversity of target types and distributions, the SAGE model can flexibly respond and always maintain high detection accuracy. It also performs well in terms of time efficiency, spending less time compared to other methods and achieving efficient and accurate object detection.

Although the SAGE model has achieved good results in the field of hyperspectral object detection, there is still room for improvement. The current model has a small amount of redundant noise in the background during detection, which is a key direction for future improvement. The follow-up research plan is to further optimize the algorithm for constructing graph structures, explore more effective feature extraction and aggregation strategies, and deeply mine pixel relationships, in order to minimize the impact of background noise on detection results and continuously improve model performance.

Reviewer #1:

The comment are as follows:

1.In Section I. INTRODUCTION, after reviewing the research status of using deep learning to construct hyperspectral object detection models, the existing problems in current research should be summarized, and a brief description should be provided on how these problems would be solved with the new proposed model.

Answer: I have reorganized the methods of detection based on the deep learning mentioned and added their shortcomings. In the last paragraph, I briefly explained how the proposed method addresses these shortcomings. The specific changes are as follows:

In recent years, deep learning has made significant progress in fields such a

---

## [Decision Letter · Decision Letter 1]

13 Feb 2025

Hyperspectral target detection based on graph sampling and aggregation network

PONE-D-24-42135R1

Dear Dr. jin,

We’re pleased to inform you that your manuscript has been judged scientifically suitable for publication and will be formally accepted for publication once it meets all outstanding technical requirements.

Kind regards,

Bardia Yousefi, Ph.D.

Academic Editor

PLOS ONE

Additional Editor Comments (optional):

Authors responded well to to the comments received.

Reviewers' comments:

Reviewer's Responses to Questions

**Comments to the Author**

1. If the authors have adequately addressed your comments raised in a previous round of review and you feel that this manuscript is now acceptable for publication, you may indicate that here to bypass the “Comments to the Author” section, enter your conflict of interest statement in the “Confidential to Editor” section, and submit your "Accept" recommendation.

Reviewer #1: All comments have been addressed

Reviewer #2: All comments have been addressed

2. Is the manuscript technically sound, and do the data support the conclusions?

Reviewer #1: Yes

Reviewer #2: Yes

3. Has the statistical analysis been performed appropriately and rigorously? 

Reviewer #1: Yes

Reviewer #2: Yes

4. Have the authors made all data underlying the findings in their manuscript fully available?

Reviewer #1: Yes

Reviewer #2: Yes

5. Is the manuscript presented in an intelligible fashion and written in standard English?

Reviewer #1: Yes

Reviewer #2: Yes

6. Review Comments to the Author

Reviewer #1: The authors have already made revisions and supplements regarding the issues that the reviewers were concerned about.

Reviewer #2: The manuscript is technically sound, with data and analyses that support its conclusions. The authors have adhered to the PLOS Data Policy by making all underlying data fully available, ensuring transparency. The writing is clear and intelligible, with only minor errors that can be addressed during copyediting.

7. PLOS authors have the option to publish the peer review history of their article (what does this mean?). If published, this will include your full peer review and any attached files.

Reviewer #1: No

Reviewer #2: No

---

## [Editor Report · Acceptance letter]

PONE-D-24-42135R1

PLOS ONE

Dear Dr. jin,

I'm pleased to inform you that your manuscript has been deemed suitable for publication in PLOS ONE. Congratulations! Your manuscript is now being handed over to our production team.

Kind regards,

on behalf of

Dr. Bardia Yousefi

Academic Editor

PLOS ONE